# Carbonic Anhydrase Inhibitors of Different Structures Dilate Pre-Contracted Porcine Retinal Arteries

**DOI:** 10.3390/ijms20030467

**Published:** 2019-01-22

**Authors:** Thor Eysteinsson, Hrönn Gudmundsdottir, Arnar Oessur Hardarson, Emanuela Berrino, Silvia Selleri, Claudiu T. Supuran, Fabrizio Carta

**Affiliations:** 1Department of Physiology, BioMedical Center, Faculty of Medicine, University of Iceland, 107 Reykjavik, Iceland; hronngu13@gmail.com (H.G.); aohardarson@gmail.com (A.O.H.); 2NEUROFARBA Department, University of Florence, Sezione di Scienze Farmaceutiche e Nutraceutiche, Via Ugo Schiff 6, 50019 Sesto Fiorentino (Florence), Italy; emanuela.berrino@unifi.it (E.B.); silvia.selleri@unifi.it (S.S.); claudiu.supuran@unifi.it (C.T.S.); fabrizio.carta@unifi.it (F.C.)

**Keywords:** carbonic anhydrase, vasodilation, vascular tone

## Abstract

Carbonic anhydrase inhibitors (CAIs), such as dorzolamide (DZA), are used as anti-glaucoma drugs to lower intraocular pressure, but it has been found that some of these drugs act as vasodilators of retinal arteries. The exact mechanism behind the vasodilatory effect is not yet clear. Here we have addressed the issue by using small vessel myography to examine the effect of CAIs of the sulfonamide and coumarin type on the wall tension in isolated segments of porcine retinal arteries. Vessels were pre-contracted by the prostaglandin analog U-46619, and CAIs with varying affinity for five different carbonic anhydrase (CA) isoenzymes found in human tissue tested. We found that all compounds tested cause a vasodilation of pre-contracted retinal arteries, but with varying efficacy, as indicated by the calculated mean EC_50_ of each compound, ranging from 4.12 µM to 0.86 mM. All compounds had a lower mean EC_50_ compared to DZA. The dilation induced by benzolamide (BZA) and DZA was additive, suggesting that they may act on separate mechanisms. No clear pattern in efficacy and affinity for CA isoenzymes could be discerned from the results, although Compound **5**, with a low affinity for all isoenzymes except the human (h) CA isoform IV, had the greatest potency, with the lowest EC_50_ and inducing the most rapid and profound dilation of the vessels. The results suggest that more than one isozyme of CA is involved in mediating its role in controlling vascular tone in retinal arteries, with a probable crucial role played by the membrane-bound isoform CA IV.

## 1. Introduction

Carbonic anhydrases (CAs; EC 4.2.1.1) are a class of ubiquitous zinc enzymes that are present in a wide variety of tissues in living organisms [1]. Their main function is to reversibly catalyze a basic physiological reaction, namely, the conversion of CO_2_ to bicarbonate and protons. These proteins are encoded by seven evolutionarily unrelated gene families, and one of these, the α-Cas, are present in vertebrates including mammals, but also in bacteria, algae, protozoans, fungi and green plants. About 16 α-CAs or CA-related proteins have been described in mammals [1,2,3], and these are known to have different tissue distribution, subcellular localization and catalytic activity, and it is unlikely that this family of CAs will be found to be larger. Among the catalytically active human expressed isoenzymes, five are expressed in the cytosol (I-III, VII and XIII), two are mitochondrial (VA and VB) one is secreted (VI), and four are transmembrane proteins (IV, IX, XII and XIV) [1,2,3]. Development of carbonic anhydrase inhibitors (CAIs) as therapeutic agents must take into consideration the fact that their effects on non-target isoenzymes must be minimal. CAIs have been developed for the treatment of a variety of diseases, since CA plays a role in several important physiological functions, such as transport of CO_2_ and bicarbonate, lipogenesis, gluconeogenesis, calcification, bone resorption, and tumorigenicity, to name a few [3]. The CA isoenzymes involved in these physiological processes are, therefore, potential therapeutic targets for treating various disorders such as cancer, epilepsy, glaucoma, acute mountain sickness, pain and osteoporosis by inhibition of target CA isozymes [4,5,6,7]. The use of CAIs for treatment of glaucoma has been well established for decades, as such drugs lower intraocular pressure (IOP) [8,9,10,11]. However, topical DRZ eye drops were found to not only lower IOP but also to increase blood flow in the retinal vasculature and optic nerve region in patients [12,13]. Similar effects of systemic applications of CAIs were found on the retinal vasculature and optic nerve oxygenation in pig eyes [14,15]. These findings suggest that as a glaucoma treatment, CAIs may have dual beneficial effects on the viability of optic nerve fibers in glaucoma, that is, they cause an increase in blood flow and oxygenation in the optic nerve region of patients, in addition to lowering their elevated IOP. Subsequently, it was found that the CAIs used for glaucoma treatment have a direct vasodilatory effect on pre-contracted retinal arteries isolated from both bovine [16] and porcine [17,18] eyes, measured by small vessel myography. The exact physiological mechanism behind the vasodilatory effect of CAIs remains unclear, although several factors, such as alterations in pH and CO_2_ levels, or processes other than CA inhibition, have been proposed [18,19]. In this study, we report the effects of new compounds with CA inhibitory properties, but different affinities for both cytosolic and transmembrane CA isoenzymes on the wall tension of isolated segments of porcine retinal arteries, measured by small vessel myography. The purpose of this study was to ascertain if one or more of the CA isozymes are involved in controlling vascular tone in these vessels, and thus shed further light on the mechanisms involved, and its location in cells.

## 2. Results

For the purposes of this study, we considered two main CA inhibitory moieties: i) The classical sulfonamide-based CAIs which include the clinically used BZA 1, DZA 2 and the synthesized benzenesulfonamide derivative **3**; and ii) the coumarin based CAIs **4**–**6** (Figure 1). The synthesis of compounds **3**–**6** was accomplished according to the procedures reported in the literature [20,21,22,23,24,25].

The compounds in Figure 1 were all tested in vitro for their inhibition potencies against the human expressed CA isoforms considered relevant in controlling vascular tone (i.e. I, II, IV, IX and XII). The results are reported in Table 1, in comparison with the standard sulfonamide inhibitor acetazolamide (AAZ). 

The data reported in Table 1 show that the BZA was a potent inhibitor against all the hCA isoforms considered here, with K_I_s in the low nanomolar concentrations. The only exception was the tumor associated isoform, IX, which was inhibited from BZA at a concentration of 45 nM. The anti-glaucoma CAI, DZA 2, showed close matching inhibition potencies to **1**, the only differences being the ineffectiveness on hCA isoforms I and IV (K_I_ values of >10000 and 8500 nM, respectively). The synthesized sulfonamide compound **3** showed a better inhibition profile when compared to the clinically used BZA and DZA, being particularly potent in inhibiting hCA II and XII, with K_I_ values of 5.8 and 9.1 nM, respectively. Higher K_I_ values were obtained for the isoform hCA IX (55.7 nM), followed by the membrane associated hCA IV (370.8 nM), and then by the ubiquitous hCA I (4014 nM). A different inhibition profile was obtained for the coumarin based scaffolds **4**–**6**. In agreement with the coumarins inhibition data reported in the literature [24,25], compounds **4**–**6** were ineffective in inhibiting the cytosolic hCA I and II (K_I_s >10000) and maintained inhibition potencies against the membrane associated ones. The 6-substituted alkyne derivative **4** showed comparable K_I_s values for hCAs IV and XII, comprised in the medium nanomolar range (69.2 and 56.4 nM, respectively), whereas the tumor associated isoform IX was inhibited at a micromolar concentration (K_I_ of 4747.6 nM). Interestingly, the 7-substituted alkyne derivative **5** retained the inhibition potency against the hCA IV with comparable K_I_ values to its regioisomer compound **4** (K_I_ of 76.4 nM). Additionally, **5** was 3.5-fold more potent than **4** in inhibiting hCA IX, but was 12.9 times less effective against hCA XII (K_I_ 730.0 nM). The 4-methyl-7-alkyl substituted coumarin **6** showed an inhibition profile close to **4**, but was by far the most potent in inhibiting all the membrane associated isoforms, that is, hCA IV, IX and XII (K_I_s of 59.7, 474.3 and 44.5 respectively). 

The sulfonamide CAI BZA 1 (Figure 1), in consideration of its enhanced acidic properties at physiological pH, has for a long time been assumed to act primarily on CAs on the extracellular side of cell membranes [27]. Such a chemical feature, however, does not significantly alter the ability of BZA to cross bio-membranes, as it has been demonstrated to accumulate in the cytoplasm of red blood cells incubating with the drug for more than 15 min [6]. We therefore examined the effect of BZA on the wall tension of porcine retinal arteries pre-contracted with 10^−6^ M of U-46619, and a representative experiment is shown in Figure 2A.

Figure 2A shows a continuous recording of wall tension of the arterial wall (in mN/mm of wall length) after normalization of contractile tension in the vessel segment. U-46619 was added to the tissue bath containing the retinal artery at the point indicated by the vertical arrow on the left. The wall tension increased from 0.4 mN/mm at the baseline to a peak force of about 1.75 mN/mm after addition of U-46619, which is comparable to the mean change in wall tension induced by the drug in healthy porcine retinal arterioles under the present experimental conditions. When wall tension had reached a peak after the addition of 10^−6^ M U-46619, 10^−3^ M BZA was added to the bath at the point indicated by the vertical arrow on the right in Figure 2A, producing an almost complete inhibition of the U-46619 induced vasoconstriction. In another experiment illustrated in Figure 2B, when 10^−3^ M DZA was added to the bath after the U-46619 reached a peak, it exerted a 33% vasodilation. When the dilation had reached a steady level, 10^−3^ M BZA was added to the bathing solution, and it immediately induced an additional dilation, to a final 100% of the maximum contraction. The concentration dependency of both compounds was examined on five different vessel segments, as shown in Figure 2C. The mean EC_50_ of BZA was calculated by best fit to the concentration-response curves as 8.43 × 10^−5^ M, and the mean EC_50_ of DZA as 8.6 × 10^−4^ M, a statistically significant difference (Two-tailed *t*-test, *p* < 0.001). Thus, BZA is a more potent vasodilator of porcine retinal arteries than DZA, as further indicated by Figure 2D, which shows the mean vasodilation induced by the compounds with a 10^−3^ M dose, expressed as a percentage of the vasoconstriction evoked by U-46619. At this concentration, BZA caused a significant mean dilatation of 78.4% ± 8.5% (n = 7, *p* < 0.002), while the mean dilation induced by 10^−3^ M **DZA** was 69.7% ± 5.7% (n = 7, *p* < 0.002). The difference in the vasodilation induced by the two compounds at 10^−3^ M was not significant (*p* = 0.398). 

The effect of CAIs listed as Compounds **3** and **4** in Table 1 on the wall tension of retinal artery segments pre-contracted with U-46619 were then examined and prepared as described in Section 4.1 on chemistry. The two compounds vary greatly in their affinity for the isoenzymes hCA II and hCA IX, as indicated in Table 1. Figure 3A shows a continuous recording of wall tension in a retinal artery segment. 

With a baseline wall tension stable at 0.6 mN/mm in Figure 3A, a dose of 10^−6^ M U-46619 was added to the bath at the point indicated by the vertical arrow. When the vasoconstriction that followed had reached a peak value of 2.4 mN/mm, a single dose of 10^−3^ M of compound **3** was added to the bathing medium. Compound **3** induced a rapid vasodilation, which in this experiment was 91% of the maximum constriction. Figure 3B presents a recording from another retinal artery segment, with the mean wall tension baseline stable at about 0.38 mN/mm. A dose of 10^−6^ M U-46619 was added and induced a peak constriction of 1.63 mN/mm. A single dose of 10^−3^ M of compound **4** was then added, and it induced a two-phased dilation in this vessel, initially a very fast one to about 0.66 mN/mm, followed by a slower phase that reached a wall tension of about 0.36 mN/mm, or an overall vasodilation of 101% in this particular experiment. The concentration dependency of both compounds **3** and **4** was examined on four different vessel segments, as shown in Figure 3C. The mean EC_50_ of compound **3** was calculated as 1.88 × 10^−4^ M, and the mean EC_50_ of compound **4** as 1.34 × 10^−4^ M, with a statistically significant difference between those two means (Two-tailed t-test, *p* < 0.02). Figure 3D compares the mean vasodilation induced by the two compounds with a 10^−3^ M dose, expressed as percentage of the vasoconstriction evoked by U-46619. Compound **3** at 10^−3^ M induced a significant mean dilatation of 102.2% ± 3.7% (n = 5, *p* < 0.01), while the mean dilation induced by 10^−3^ M of compound **4** was 98.8% ± 6.1% (n = 4, *p* < 0.002). The difference in vasodilation induced by the two compounds at 10^−3^ M was not significant (*p* = 0.343). 

The effect of two new coumarin CAIs listed as compounds **5** and **6** in Table 1—both with low affinity for human CA isoenzymes I and II—on arterial wall tension was then tested in the same manner as for compounds **3** and **4**, and the results are illustrated in Figure 4. An experiment examining the effect of a single dose of 10^−3^ M of compound 5 is shown in Figure 4A. 

In Figure 4A, with the baseline wall tension stable at 1.54 mN/mm, 10^−6^ M U-46619 was added to the bath at the point indicated by the vertical arrow. When the vasoconstriction induced by the drug had reached a peak value of 3.38 mN/mm, 10^−3^ M of compound **5** was added to the bath at the time indicated by the right vertical arrow. Compound **5** evoked a rapid vasodilation, which reached a level that was lower than the baseline wall tension before addition of U-46619, or 0.77 mN/mm, that is, the vasodilation in the experiment illustrated reached 140% of the maximum contraction of the vessel at peak. Figure 4B presents a recording from another retinal artery segment, with the mean wall tension baseline reaching a stable level of 0.95 mN/mm. A dose of 10^−6^ M U-46619 was added and induced a peak constriction of 2.21 mN/mm. A single dose of 10^−3^ M compound **6** was then added and induced a slow vasodilation, which was 96.8% of the maximum contraction of the vessel. The concentration dependency of compounds 5 and 6 was examined on four different vessel segments, as shown in Figure 4C. The mean EC_50_ of compound **5** was calculated in the same manner as 4.12 × 10^−6^ M, and the mean EC_50_ of compound **6** as 1.01 × 10^−4^ M, a statistically significant difference (Two-tailed *t*-test, *p* < 0.001). The mean vasodilation induced by the two compounds with a 10^−3^ M dose, expressed as a percentage of the vasoconstriction evoked by U-46619, is illustrated in Figure 4D. At 10^−3^ M, compound 5 induced a significant mean dilation of 116.5% ± 1.7% (*n* = 4, *p* < 0.001), while the mean dilation induced by 10^−3^ M of compound **6** was 99.5% ± 1.2% (*n* = 4, *p* < 0.002). The difference in the vasodilation induced by the two compounds at 10^−3^ M was significant (Two tailed *t*-test, *p* < 0.001).

## 3. Discussion

The present study shows for the first time that BZA causes a vasodilation in isolated porcine retinal arterioles, which is significantly more profound than that induced by DZA, as found previously by some studies [16,18]. We are not aware of any previous study showing a direct vasodilatory effect on isolated vessels by this compound. The results support the suggestion that BZA is a membrane permeable CAI, as has been shown in isolated red blood cells [6], although it has so far been generally (and erroneously) assumed to be an impermeant CAI and has been used as such [28,29,30,31]. The dilation induced by BZA may be due to inhibition of one or more of either cytosolic- or membrane-bound CA isoenzymes, or both. The dilating effect of the compound is additive to that induced by DZA, suggesting that the vasodilation induced by these compounds may involve separate mechanisms to some extent. BZA has been proposed as a putative treatment for acute mountain sickness [7] and myocardial ischemia [32], primarily due to its effect on acidity, but the potent effect on vascular tone demonstrated here raises the possibility of other putative therapeutic uses. On a cellular level, BZA has been found to inhibit calcium currents mediated by voltage-sensitive calcium channels expressed on the membranes of HEK293 cells [33], and low-threshold calcium currents in hippocampal pyramidal cells [34]. It is not clear if these effects are related to the vasodilatory effects of BZA and other CAIs shown here, but it has been shown that charybdotoxin, the selective blocker of calcium activated potassium channels (K_Ca_), blocks vasodilation of isolated mesenteric arteries induced by CAIs [35]. The present results also suggest that more than one isoenzyme of CA is likely to be involved in mediating the vasodilation, and thus in maintaining vascular tone. The physiological mechanisms involved, aside from CA inhibition, are still unclear, although several hypotheses have been formulated. It has been found that changes in pH or CO_2_ levels induced by CA inhibition are not the mechanisms responsible for the vasodilation, since vasodilation can still be induced by CAIs at normal pH, but acidosis does appear to reduce the effect to some extent [16,18]. Intracellular acidosis in vascular smooth muscle cells is increased by DZA, but that increase is not related to relaxation [19]. Hypercapnia, however, has no effect on the vasodilatory effect of CAIs [18]. The perivascular tissue plays a role, since vasodilation is significantly reduced in isolated porcine retinal arterioles, with perivascular tissue removed [17]. There is evidence that nitric oxide (NO), presumably released from the endothelium, is involved in the vasodilatory effect of CAI, since it is reduced in the presence of nitric oxide synthase (NOS) inhibitor L-NAME or guanylyl cyclase inhibitor ODQ [19,36]. It has also been argued that inhibition of CA is not necessarily the mechanism responsible for vasodilatory effects of the inhibitors, due to the relatively high concentrations needed [18,37], but we have shown here that some CAIs, such as compound 5 in the present study, induce vasodilation at a low concentration, with a low, micromolar EC_50_. In this study, we have examined the vasodilatory effects of a series of sulfonamide/coumarin CAIs with varying structure and affinity for different isoenzymes of CA, and find that all of them, regardless of selectivity, induced vasodilation in pre-contracted retinal arteries, but with some differences in potency, which may be related to their relative affinity for the CA isozymes. Further studies of the effect of CAIs with different selectivity for cytosolic versus membrane bound isozymes, and greater specificity for individual isozymes are needed to clarify the role of carbonic anhydrase in controlling vascular tone in retinal arteries.

## 4. Materials and Methods 

### 4.1. Chemistry

Anhydrous solvents and all reagents were purchased from Sigma-Aldrich (Milan, Italy), Alfa Aesar (Milan, Italy) and TCI (Milan, Italy). All reactions involving air- or moisture-sensitive compounds were performed under a nitrogen atmosphere using dried glassware and syringes techniques to transfer solutions. Nuclear magnetic resonance (^1^H-NMR, ^13^C-NMR) spectra were recorded using a Bruker Avance III 400 MHz spectrometer in DMSO-*d_6_*. Chemical shifts are reported in parts per million (ppm) and the coupling constants (*J*) are expressed in Hertz (Hz). Splitting patterns are designated as follows: s, singlet; d, doublet; t, triplet; q, quadruplet; m, multiplet; brs, broad singlet; dd, double of doublets. The assignment of exchangeable protons (OH and NH) was confirmed by the addition of D_2_O. Analytical thin-layer chromatography (TLC) was carried out on Merck silica gel F-254 plates. Melting points (m.p.) were carried out in open capillary tubes and are uncorrected. The solvents used in MS measures were acetone, acetonitrile (Chromasolv grade), purchased from Sigma–Aldrich (Milan, Italy), and mQ water 18 MX, obtained from Millipore’s Simplicity system (Milan, Italy). The mass spectra were obtained using a Varian 1200L triple quadrupole system (Palo Alto, CA, USA) equipped by Electrospray Source (ESI), operating in both positive and negative ions. Stock solutions of analytes were prepared in acetone at 1.0 mg mL^−1^ and stored at 4 °C. Working solutions of each analyte were freshly prepared by diluting stock solutions in a mixture of mQ H_2_O/ACN 1:1 (*v/v*) up to a concentration of 1.0 μg mL^−1^. The mass spectra of each analyte were acquired by introducing via syringe pump at 10 μL min^−1^ its working solution. Raw data were collected and processed by Varian Workstation Vers. 6.8 software.

### 4.2. Carbonic Anhydrase Inhibition

An Applied Photophysics stopped-flow instrument was used for assaying the CA catalyzed CO_2_ hydration activity [1]. Phenol red (at a concentration of 0.2 mM) was used as an indicator, working at an absorbance maximum of 557 nm, with 20 mMHepes (pH 7.5) as buffer, and 20 mM Na_2_SO_4_ for maintaining constant ionic strength, following the initial rates of CA-catalyzed CO_2_ hydration reaction for a period of 10–100 s. CO_2_ concentrations ranged from 1.7 to 17 mM for the determination of the kinetic parameters and inhibition constants. For each inhibitor, at least six traces of the initial 5%–10% of the reaction were used to determine the initial velocity. The uncatalyzed rates were determined in the same manner and subtracted from the total observed rates. Stock solutions of inhibitor (0.1 mM) were prepared in distilled and deionized water, and dilutions up to 0.01 nM were carried out with the assay buffer. Inhibitor and enzyme solutions were preincubated together for 15 min at room temperature prior to assay, to allow for the formation of the E-I complex. The inhibition constants were obtained by non-linear least-squares methods using PRISM 3 and the Cheng–Prusoff equation, as reported earlier [24,25]; these represent the mean from at least three different determinations. All CA isoforms were recombinant ones obtained in-house, as reported previously [24,25].

### 4.3. Myography Preparation

Changes in the wall tension of short segments (<2 mm) of porcine retinal arterioles were examined with a small vessel myography system. Pig eyes were obtained from a local abattoir. The pigs were anesthetized with CO_2_ and put down by exsanguination. All procedures involving animals adhered to the appropriate local and international laws and relevant ethical rules. Only one eye was obtained from each animal for the experiments. Eyes were placed in a 4 °C oxygenated physiological saline solution (PSS) and transported to the laboratory as quickly as possible (within 30 min). The composition of PSS used for both transport of the eyes and as the extracellular bathing solution for myography recordings from isolated retinal arterioles was as follows (in mM): 112.6 NaCl; 5.91 KCl; 24.9 NaHCO_3_; 1.19 MgCl_2_; 1.18 NaH_2_PO_4_; 2.0 CaCl_2_; 11.5 glucose, all dissolved in double distilled water. The PSS was oxygenated by a mixture of 95% O_2_ and 5% CO_2_ with pH maintained at 7.4.

The eyes were bisected with a razor blade at the equator and the anterior segment removed. The vitreous was then removed from the posterior segment and replaced with 4 °C oxygenated PSS. The posterior segment containing the optic disk was then placed under a stereoscope. The arteries were identified based on their smaller diameter compared to the adjacent venule, and the color of the blood inside the vessels. A straight arteriolar segment was then selected as close to the optic disc as possible, and an approximately 2 mm long segment was dissected, with retinal tissue extending approximately 1 mm on either side of the vessel. Dissected segments of porcine retinal arteries were mounted in a DMT630MA wire myograph system (DMT A/S, Aarhus, Denmark) for measurement of contractile activity. The myograph system consisted of four separate tissue baths, so that four vessel segments could be examined simultaneously, with a volume of up to 10 ml, and in each bath there is a force transducer to measure tension to a thin tungsten wire, placed in the lumen of a vessel. The recording chambers can be automatically emptied of PSS with suction and quickly refilled again with an automatic buffer filler system (625FS, DMT A/S, Aarhus, Denmark). The vessel segments were placed in between two metal jaws in the chamber, with two screws on the jaws to attach a tungsten wire to it. The arteries were mounted on two tungsten wires with a diameter of 25 µm, guided by a stereo microscope. Two incisions were made through the retinal artery and adjacent retina in situ, and one tungsten wire guided first through the lumen and along the vessel from one incision to the other. Incisions on both sides of the arterial segment through the retina were then made, and the retinal vessel segment, with the wire inside the lumen, transferred to the myography recording chamber. The wire was first tied with the screws on both ends to the jaw attached to the micrometer. A second wire was then guided through the lumen along the top of the first wire, and then attached to the jaw of the force transducer with the screws there. The exact length of the vessel segment, once placed in the myography system, was measured by taking a photograph of it through the stereoscope, and then converting the vessel length in pixels into millimeters. The heating unit of the DMT630MA system was then turned on and set to a stable temperature of 37 °C, monitored by a thermal probe inserted into the bathing medium.

### 4.4. Myography Procedure

The wall tension of the vessel segment was continuously recorded with LabChart Pro (ADInstruments, Oxford, UK). Once the temperature in the bathing medium had reached 37 °C, the preparation was left to stabilize for 30 min and the normalization of wall tension was carried out, taking into account the measured length of the vessel. The normalization was done so that results from vessel segments with different diameters were comparable. The procedure involves increasing the vessel diameter by separating the wires inside the lumen by the micrometer in fixed steps, and at each step determine the passive tension of the vessel wall (which corresponds to a transmural pressure between about 0–70 mmHg). The diameter-tension relationship in this is exponential, and was extrapolated from the measurements using the DMT Normalization module of LabChart Pro. The intercept between the exponential function of measured diameter-tension and a straight line was calculated, based on the Laplace equation (wall tension = transmural pressure × radius) with transmural pressure in the equation set to 70 mmHg. By adjusting the built-in micrometer in each chamber of the myograph system, the jaws were set so that they were placed at about 94% of the intercept length, as calculated by the DMT Normalization module. The wall tension at that level is at maximum tension (i.e., the optimal passive stretch of the smooth muscle cells of the vessel wall).

After normalization the vessel segments were pre-contracted with the thromboxane A2 analogue U-46619 (9,11-dideoxy-9 α,11α methanoepoxy prostaglandin F_2_α) (Cayman Chemicals Inc., Tallinn, Estonia), which was dissolved in demineralized water to a stock solution of 10^−3^ M. The stock solution was added to the myography tissue bath to reach a final concentration of 10^−6^ M. Experiments were discarded if the peak increase in wall tension induced by 10^−6^ M U-46619 was less than 0.5 mN/mm. Once the vasoconstriction induced by U-46619 reached a peak, the CAI to be tested was added to the bathing medium. Stock solutions of all carbonic anhydrase inhibitors were made at 0.1 M concentration, dissolved in DMSO, and then diluted to the desired concentration. A dose of 10^−3^ M of all the CAIs examined in this study was in each case tested first, as a point of reference to ascertain if the compound had any effect on wall tension of the arteries.

Concentration-response curves were obtained by adding the lowest concentration tested, in most cases 10^−6^ M or lower, to the bathing medium and then adding the next step (e.g., 2 × 10^−6^ M) when the change in wall tension induced by the previous concentration had reached a steady level.

### 4.5. Data Analysis

The myography recording traces were copied from LabChart Pro to SigmaPlot 13 (Systat Software Inc, San Jose, CA, USA) for graphic analysis and presentation, as well as all other graphics. Statistical analysis was performed using Systat (Systat Software Inc, San Jose, CA, USA). The estimation of EC_50_ from normalized concentration-response curves was calculated by best fit, using GraphPad Prism 8 (GraphPad Software Inc., San Diego, CA, USA). The results are expressed as means ± SEM. 

## Figures and Tables

**Figure 1 ijms-20-00467-f001:**
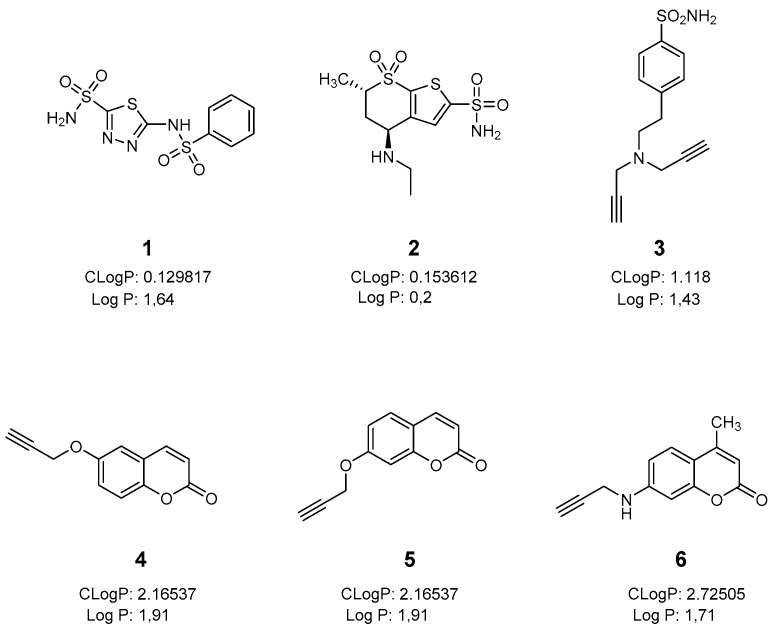
Chemical structures of compounds **1**–**6** used in this study.

**Figure 2 ijms-20-00467-f002:**
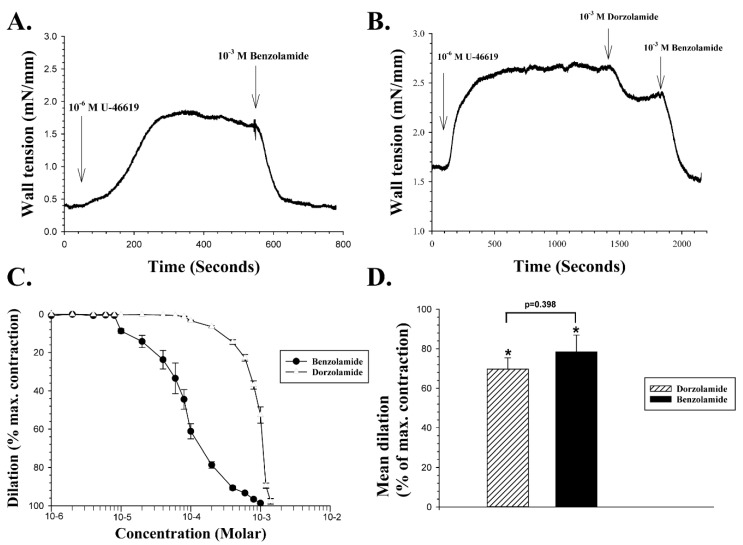
The effects of U-46619, BZA and DZA on wall tension in porcine retinal arteries. **A**: Continuous myography recording of wall tension in one artery. The vertical arrow on the left indicates the time point when 10^−6^ M U-46619 was added to the bath. The wall tension then increases and reaches a peak, and 10^−3^ M BZA was added at the time indicated by the right vertical arrow. Note the rapid vasodilation induced by BZA to the baseline level. **B**: Myography recording from another artery. After vasoconstriction induced by U-46619 reached a peak, 10^−3^ M DZA was added at the time indicated by the middle arrow. The left vertical arrow indicates an addition of 10^−3^ M BZA. Note the additional dilation induced by BZA. **C**: Mean concentration-response curves for the dilatory effects of BZA (filled circles) and a (open triangles), expressed as a percentage of the maximum contraction induced by U-46619. Symbols represent the mean ± standard error of the mean (SEM) dilation at each concentration. **D:** The mean ± SEM dilation induced by BZA and DZA in response to a single dose of 10^−3^ M added to the bathing solution, as a percentage of the maximum contraction induced by U-46619. An asterix (*) above bars indicates statistically significant vasodilation (*p* < 0.002)

**Figure 3 ijms-20-00467-f003:**
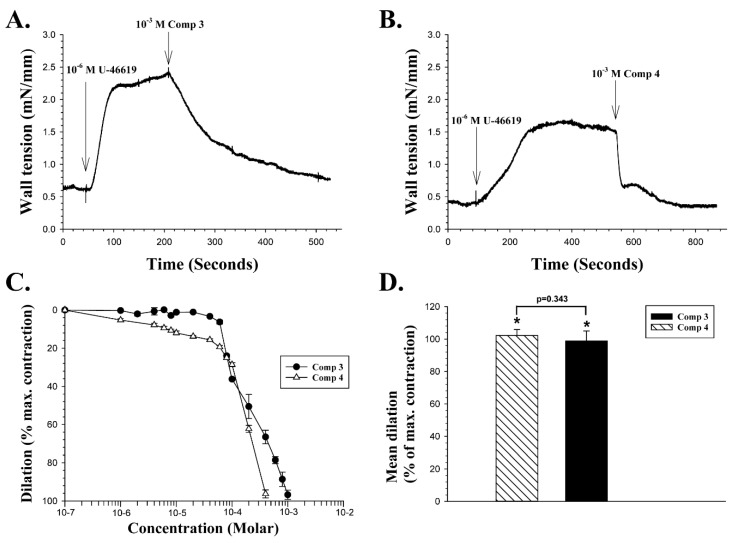
The effects of U-46619 and compounds **3** and **4** on retinal arterial wall tension. **A**: Continuous recording of wall tension. First, 10^−6^ M U-46619 was added to the bath, and when the vasoconstriction reached a peak in the experiment shown, a single dose of 10^−3^ M of compound **3** was added at the point indicated by the right vertical arrow. **B**: Continuous recording of wall tension from another vessel. At the peak of vasoconstriction induced by U-46619, a dose of 10^−3^ M of compound **4** was added at the point indicated by the right arrow. **C**: Mean concentration-response curves for the dilatory effects of compounds **3** (filled circles) and **4** (open triangles), expressed as a percentage of U-46619 induced maximum contraction. **D**: The mean dilation induced by compounds **3** and **4** in response to a single dose of 10^−3^ M added to the bathing solution, as a percentage of the maximum contraction induced by U-46619. An asterix (*) above a bar indicates statistically significant vasodilation (*p* < 0.01 and *p* < 0.002).

**Figure 4 ijms-20-00467-f004:**
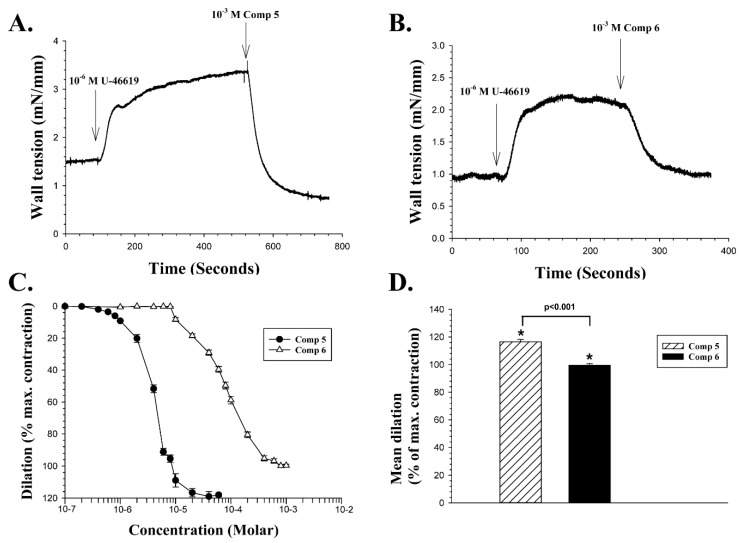
The effects of U-46619, and compounds **5** and **6** on retinal arterial wall tension. **A**: Continuous recording of wall tension. After adding 10^−6^ M U-46619 to the bath, and when constriction had reached a peak, a single dose of 10^−3^ M of compound **5** was added, as indicated by the right vertical arrow. **B**: Recording of wall tension from a second vessel. At the peak of vasoconstriction induced by U-46619, a dose of 10^−3^ M of compound **6** was added at the point indicated by the right arrow. **C**: Mean concentration-response curves for the dilatory effects of compounds **5** (filled circles) and **6** (open triangles), expressed as a percentage of U-46619 induced maximum contraction. **D**: The mean dilation induced by compounds **5** and **6** in response to a single dose of 10^−3^ M added to the bathing solution, as a percentage of the maximum contraction induced by U-46619. Asterix (*) above the bars indicate statistically significant vasodilation (*p* < 0.001 and *p* < 0.002).

**Table 1 ijms-20-00467-t001:** Inhibition data of human CA isoforms I, II, IV, IX and XII with compounds **1**–**6** and AAZ by a stopped flow CO_2_ hydrase assay [26].

K_I_ (nM) *					
Cmp	hCA I	hCA II	hCA IV	hCA IX	hCAXII
**1 (BZA)**	15.0 [1]	9.0	12.0	45.0	3.5 [2]
**2 (DZA)**	>10000 [3]	9.0	8500	52.0	3.5
**3**	4014.0	5.8	370.8	55.7	9.1
**4**	>10000	>10000	69.2	4747.6	56.4
**5**	>10000 [4]	>10000 [4]	76.4	1350.0 [4]	730.0 [4]
**6**	>10000	>10000	59.7	474.3	44.5
**AAZ**	250	12	74	25.8	5.7

* Mean from 3 different assays, by a stopped flow technique (errors were in the range of ± 5%–10% of the reported values).

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
