# Peer review of "Carbonic Anhydrase Inhibitors of Different Structures Dilate Pre-Contracted Porcine Retinal Arteries"

_ijms, 2019, doi:10.3390/ijms20030467_

Reviewer 1 Report

The work of Eysteinsson and colleagues described the synthesis and biological investigation of a number of CA inhibitors from two different chemical and pharmacophoric classes as vasodilators in pre-contracted porcine retinal arteries. The work is sound, based on solid background, and generally well written. Results are of great interest given the new findings provided by the authors, namely that CA inhibition is a new mechanism of vasodilation in isolated vessels. Although for some of tested compounds CA inhibition is not the only mechanism potentially related to the vasodilatatory effect, this results will trigger additional investigations. 

Overall, is the opinion of this referee that the work deserves publication in the journal, pending a few minor edits to the text:

1) Authors should use subscripts for EC50 also in the abstract.

2) Among the wide panel of CA inhibitors available in authors' labs and in the literature, why authors have selected Compounds 1-6 for this study? It is curious to note that most of them have an alkyn moiety.

Author Response

1) Authors should use subscripts for EC50 also in the abstract.

Our response: this has been corrected.

2) Among the wide panel of CA inhibitors available in authors' labs and in the literature, why authors have selected Compounds 1-6 for this study? It is curious to note that most of them have an alkyn moiety.

Our response: We have altered the EC50 in the abstract as suggested.  The Table 1 in part explains the reason why these compounds were selected, as it shows their affinity for the different isoenzymes of human carbonic anhydrase, and that they differ in their selectivity.  The insertion of the terminal alkyl moiety was chosen as it determines the appropriate LogP values (i.e. close to the benzolamide) on the scaffolds selected. This does shed light on the putative role of the isoenzymes in mediating the vasodilatory effect, which is well established for dorzolamide, i.e. compound 1.  Dorzolamide was selected as a reference, benzolamide because it has been inappropriately claimed to be membrane impermeant, but its effect on vascular tone unknown, and Compounds 3-6 because of their different affinities for hCA II and hCAIV.

We apreciated indeed the suggestions and comments rose from this Reviewer

Reviewer 2 Report

The manuscript focuses on an interesting topic in the field of CA inhibition. It is aimed to investigate the effects of several carbonic anhydrase inhibitors of sulfonamide and coumarin type (showing varying affinity for five different CA human isoenzymes) on the wall tension in isolated segments of porcine retinal arteries. Collectively the results suggest the functional involvement of several CA isoforms in the vascular tone of retinal arteries with a probable  key role played by the membrane bound CA IV. However, some issues need to be addressed by the authors.

Abstract: Line 16-1: the sentence is too long

Methods: the ethical declaration statement is lacking in the method section

Methods: paragraph 4.3, line 312: the pH of the physiological saline solution has to be reported

Pag 8 line 317: “was then selected” instead of “was then be selected”

Pag 4 line 144-148: I suggest to re-write the sentences avoiding repetition and redundancy.  A possible suggestion could be the following: line 144” …10-3M BZA was added to the bath at the time point indicated by the vertical arrow on the right in Fig. 2A, producing an almost complete inhibition of the U-46619 induced contraction. When DZA was added to the bath after the vasocostriction peak induced by U-46619, it exerted a 33% inhibition, lower with respect to the effect produced by BZA…” 

Fig.2: When the author report the mean dilation values at 10-3M the differences among the two compounds (Fig. 3D) are not statistically significant, despite results reported in Fig. 2A, B and C show the contrary. The author should justify and discuss the lack of statistical significance in the mean values. Perhaps the high variability in the measures and the low number of replicates could help in the explanation of this aspect of the results. Did the authors test the additive effects of BZA with respect to DZA at concentrations different than 10-3M?

Pag 5 line 163: “Fig 3A” instead of “Fig 2A”

Pag 5 line 182: “it induced” instead of “which induced”

Pag. 5 line 186-197: the sentence “a statistically significant difference ….” is not clear.

Pag 6: line 195: “Figure 4” instead of “Figure 3”

Pag 7 Line 225 and through the text:  Why the authors indicate the statistical notation as p>0.01 and not p<****?< p="">

Pag 7 line 234: the sentence has to be re-written. Some suggestion “…of either cytosolic or membrane-bound CA isoenzymes…”

Discussion: line 236-237: as reported above the author should discuss the lack of a statistically significant difference for the mean effects of the two compounds BZA and DZA at 10-3M.

Author Response

Abstract: Line 16-1: the sentence is too long

Our response: the sentence has now been split into two shorter sentences.

Methods: the ethical declaration statement is lacking in the method section

Our response: the tissue used were obtained from a local abattoir serving the food industry, and the enucleation approved by the chief veterinary officer responsible. Thus, ethical or legal issues involving experimental animals otherwise do not apply.  However, we have added the following sentence in the methods section (page 8, lines 307-308): “All procedures involving animals adhered to the appropriate local and international laws and relevant ethical rules”.

Methods: paragraph 4.3, line 312: the pH of the physiological saline solution has to be reported

Our response: The following sentence has been added to the paragraph (line 313-314):” The PSS was oxygenated by a mixture of 95% O2 and 5% CO2 with pH maintained at 7.4“

Pag 8 line 317: “was then selected” instead of “was then be selected”

Our response: this has been altered in the revised manuscript.

Pag 4 line 144-148: I suggest to re-write the sentences avoiding repetition and redundancy.  A possible suggestion could be the following: line 144” …10-3M BZA was added to the bath at the time point indicated by the vertical arrow on the right in Fig. 2A, producing an almost complete inhibition of the U-46619 induced contraction. When DZA was added to the bath after the vasoconstriction peak induced by U-46619, it exerted a 33% inhibition, lower with respect to the effect produced by BZA…” 

Our response: we have now modified the passage along these lines suggested by the reviewer, although we don´t think the term “inhibition” is warranted here. We measure vascular tone and therefore don´t know if inhibition is involved, but we see a 33% vasodilation…

Fig.2: When the author report the mean dilation values at 10-3M the differences among the two compounds (Fig. 3D) are not statistically significant, despite results reported in Fig. 2A, B and C show the contrary. The author should justify and discuss the lack of statistical significance in the mean values. Perhaps the high variability in the measures and the low number of replicates could help in the explanation of this aspect of the results. Did the authors test the additive effects of BZA with respect to DZA at concentrations different than 10-3M?

Our response: the mean vasodilation induced by 10-3 M of compounds 3 and 4 depicted in Fig 3D was in both instances around 100%, so clearly each by itself caused a statistically significant vasodilation.  But the difference in the vasodilation induced by them was not statistically significant, so we fear there is a misunderstanding here. In some case a single dose of 10-3 M dorzolamide (DZA) induced no more than about 50% vasodilation (the mean was 69,7±5.7 per cent), as shown in Fig 2B, and if 10-3 M benzolamide (BZA) was added to that it would add to the vasodilation, so the same 10-3 M concentration of both drugs were used in the Fig. 2A, B and C. The recording depicted in Fig 2B shows a clear example of the additive effect. 

Pag 5 line 163: “Fig 3A” instead of “Fig 2A”

Our response: this has been corrected.

Pag 5 line 182: “it induced” instead of “which induced”

Our response: this has been corrected.

Pag. 5 line 186-197: the sentence “a statistically significant difference ….” is not clear.

Our response: The end of the sentence has been modified to:”… with a statistically significant difference between those two means.“

Pag 6: line 195: “Figure 4” instead of “Figure 3”

Our response: this has been corrected.

Pag 7 Line 225 and through the text:  Why the authors indicate the statistical notation as p>0.01 and not p<****?< span="">

Our response: this has been corrected everywhere in the manuscript now.

Pag 7 line 234: the sentence has to be re-written. Some suggestion “…of either cytosolic or membrane-bound CA isoenzymes…”

Our response: The sentence now reads: “The dilation induced by BZA may be due to inhibition of one or more of either cytosolic or membrane-bound CA isoenzymes, or both“.

Discussion: line 236-237: as reported above the author should discuss the lack of a statistically significant difference for the mean effects of the two compounds BZA and DZA at 10-3M.

Our response: see response on this issue above. Both compounds cause a highly significant vasodilation in response to a single dose of 10-3M. But the vasodilation induced by them is not significantly different.

We are grateful to this Reviewer for the comments and corrections suggested

Reviewer 3 Report

A brief summary

The aim of this manuscript is to test a total of 6 carbonic anhydrase inhibitors (CAIs) which have different Kis for isoforms I, II, IV, IX and XII, on their potency and efficacy in inducing vasodilation of pre-contracted porcine retinal arteries, in an attempt to deduce the specific isoform of CAs that might be the relevant targets in this phenomenon. Their results at least suggest that not a single isoform is solely responsible for the CAI-induced vasodilatory effect.

Broad comments

It is interesting and important to understand the mechanisms of vasodilation induced by CAIs for potential beneficial clinical applications. The strength of the manuscript is that the authors have well characterized CAIs with varied affinities for different isoforms of CAs including the cytosolic (CA I, II) as well as plasma membrane (CA IV, IX, XII) forms. If there is a strong correlation between the affinity of the inhibitors and the efficacy of vasodilation, the target of CAI might be deduced.

The weakness of the paper lies in the premise of this hypothesis. Torring et al (2009), using AAZ and DZA in the same in vitro model of porcine arteries, clearly demonstrated that the vasodilatory effect involved mechanisms other than CA enzymatic activities as vasodilation was independent of the substrates of CAs and likely associated with the properties of high concentrations of CAIs. Therefore, to further identify which isoform might be most responsible for the phenomenon of CAI-induced vasodilation seems less founded. In addition, a survey of the relative abundance of CA isoforms I, II, IV, IX and XII should be established via Western analysis using the experimental materials to at least provide the relevancy for the experiments.

Consequently, the claim in the Abstract “The results suggest that more than one isozyme of CA is involved in mediating its role in controlling vascular tone in retinal arteries, with a probable crucial role played by the membrane-bound isoform CA IV” cannot be substantiated, because of the lack of data stated in the above.

Specific comments

1.       There appear gross errors (not sure whether due to editorial) in stating the p values in the entire manuscript, involving lines 154, 157, 158, 187, 190, 191, 220, 223, 224, 225: all “>” should be changed to “<”. In addition, the p value in line 220 is invalid.

2.       In line 160, the description “… two new sulfonamide CAIs listed as Compounds 3 and 4 in Table 1” is inconsistent with the earlier description for these compounds in line 71 as well as the structures shown in Figure 1.

3.       For the Bar Graphs in Ds in Figures 2-4, the description of “*” should be mentioned in the legends, and another p value should be added between the two bars.

Author Response

The weakness of the paper lies in the premise of this hypothesis. Torring et al (2009), using AAZ and DZA in the same in vitro model of porcine arteries, clearly demonstrated that the vasodilatory effect involved mechanisms other than CA enzymatic activities as vasodilation was independent of the substrates of CAs and likely associated with the properties of high concentrations of CAIs. Therefore, to further identify which isoform might be most responsible for the phenomenon of CAI-induced vasodilation seems less founded. In addition, a survey of the relative abundance of CA isoforms I, II, IV, IX and XII should be established via Western analysis using the experimental materials to at least provide the relevancy for the experiments.

 Our response: we have not yet measured the expression of the CA isoforms in the porcine retinal arteries, by Western blot or any other means, but we plan to do that. So the reviewer is right that we cannot be sure about the exact isoforms involved, but it is likely that the isoforms specified in the table, being common in mammals, are involved, and therefore it is relevant to look at the effects of compounds with known different affinities for these isoforms, as we have done.  We are aware of the Torring et al (2009) study, and it is cited and addressed in the Discussion, and the other putative mechanisms that may be involved in mediating the vasodilatory effects are also specified and addressed there. Nevertheless, Torring et al could not conclude that CA inhibition or modification of CA enzymatic activity was not at least a contributing factor.  We agree with their notion that probably a cascade of events are triggered by the CAIs leading to vasodilation, and CA inhibition is possibly not the most important one.  We think that the issue is by no means settled, as we suggest in the Discussion. We intend to work further on the matter, but we still are of the opinion that the experiments and results described in our paper do indicate that more than one isoform, including CA IV are involved.

Consequently, the claim in the Abstract “The results suggest that more than one isozyme of CA is involved in mediating its role in controlling vascular tone in retinal arteries, with a probable crucial role played by the membrane-bound isoform CA IV” cannot be substantiated, because of the lack of data stated in the above.

Our response: we are basing our conclusion on the relationship between the different Kis for isoforms I, II, IV, IX and XII, on their potency and efficacy in inducing vasodilation.  Compounds with fairly selective high affinity for isoform CA IV, and very low for the others, have a potent vasodilatory effect.  We think that warrants concluding that the membrane-bound isoform may be important. But this certainly needs further study, as stated above.

Specific comments

1.       There appear gross errors (not sure whether due to editorial) in stating the p values in the entire manuscript, involving lines 154, 157, 158, 187, 190, 191, 220, 223, 224, 225: all “>” should be changed to “<”. In addition, the p value in line 220 is invalid.

Our response: this has been corrected everywhere in the manuscript now.

2.       In line 160, the description “… two new sulfonamide CAIs listed as Compounds 3 and 4 in Table 1” is inconsistent with the earlier description for these compounds in line 71 as well as the structures shown in Figure 1.

Our response: this has been corrected in the manuscript and line 160 now reads “The effect of CAIs listed as Compounds 3 and 4 in Table 1…”

3.       For the Bar Graphs in Ds in Figures 2-4, the description of “*” should be mentioned in the legends, and another p value should be added between the two bars.

Our response: this has been amended in the manuscript as suggested, and the p values added.

We are grateful to this Reviewer for the attention dedicated to our manuscript which now was greatly improved

Round  2

Reviewer 3 Report

Though technical errors were corrected in the revised manuscript, the scientific concern raised in the first reivew was not experimentally addressed which does not allow a fundamental change in the opinion of the manuscript.